# Surgical treatment of spondylolisthesis by oblique lumbar interbody fusion and transpedicular screw fixation: Comparison between conventional double position versus navigation-assisted single lateral position

Junghoon Han[1☯], Chang-Min Ha[2☯], Woon Tak Yuh[3], Young San Ko[4], Jun-Hoe Kim[1], Tae-Shin Kim[1], Chang-Hyun Lee[1,5], Sungjoon Lee[2,6], Sun-Ho Lee[2,6], Asfandyar Khan[7], Chun Kee Chung[1,5,8], Chi Heon Kim[1,5,9]*

1 Department of Neurosurgery, Seoul National University Hospital, Seoul, Republic of Korea, 2 Department of Neurosurgery, Samsung Medical Center, Seoul, Republic of Korea, 3 Department of Neurosurgery, Dongtan Sacred Heart Hospital, Hwaseong, Republic of Korea, 4 Department of Neurosurgery, Keimyung University Dongsan Hospital, Daegu, Republic of Korea, 5 Department of Neurosurgery, Seoul National University College of Medicine, Seoul, Republic of Korea, 6 Department of Neurosurgery, Sungkyunkwan University School of Medicine, Suwon-si, Gyeonggi-do, Republic of Korea, 7 School of Medicine, Newcastle University, Newcastle upon Tyne, United Kingdom, 8 Department of Brain and Cognitive Sciences, Seoul National University, Seoul, Republic of Korea, 9 Department of Medical Device Development, Seoul National University College of Medicine, Seoul, Republic of Korea

☯ These authors contributed equally to this work.
* chiheon1@snu.ac.kr

## Abstract

### Background and objectives

Oblique lumbar interbody fusion (OLIF) procedures involve anterior insertion of interbody cage in lateral position. Following OLIF, insertion of pedicle screws and rod system is performed in a prone position (OLIF-con). The location of the cage is important for restoration of lumbar lordosis and indirect decompression. However, inserting the cage at the desired location is difficult without reduction of spondylolisthesis, and reduction after insertion of interbody cage may limit the amount of reduction. Recent introduction of spinal navigation enabled both surgical procedures in one lateral position (OLIF-one). Therefore, reduction of spondylolisthesis can be performed prior to insertion of interbody cage. The objective of this study was to compare the reduction of spondylolisthesis and the placement of cage between OLIF-one and OLIF-con.

### Methods

We retrospectively reviewed 72 consecutive patients with spondylolisthesis for this study; 30 patients underwent OLIF-one and 42 underwent OLIF-con. Spinal navigation system was used for OLIF-one. In OLIF-one, the interbody cage was inserted after reducing spondylolisthesis, whereas in OLIF-con, the cage was inserted before reduction. The following parameters were measured on X-rays: pre- and postoperative spondylolisthesis slippage, reduction degree, and the location of the cage in the disc space.

**Data Availability Statement:** All relevant data are within the paper and its Supporting Information files.

**Funding:** This study was supported by Seoul National University Hospital research fund (grant no. 04-2021-0540); and by Doosan Yonkang Foundation (800-20210527). But the funders had no role in study design, data collection and analysis, decision to publish, or preparation of the manuscript.

**Competing interests:** The authors have declared that no competing interests exist.

## Results

Both groups showed significant improvement in back and leg pains ($p < .05$). Transient motor or sensory changes occurred in three patients after OLIF-con and in two patients after OLIF-one. Pre- and postoperative slips were 26.3±7.7% and 6.6±6.2% in OLIF-one, and 23.1±7.0% and 7.4±5.8% in OLIF-con. The reduction of slippage was 74.4±6.3% after OLIF-one and 65.4±5.7% after OLIF-con, with a significant difference between the two groups ($p = .04$). The cage was located at 34.2±8.9% after OLIF-one and at 42.8±10.3% after OLIF-con, with a significant difference between the two groups ($p = .004$).

## Conclusion

Switching the sequence of surgical procedures with OLIF-one facilitated both the reduction of spondylolisthesis and the placement of the cage at the desired location.

## Introduction

Lumbar spondylolisthesis is a common cause of spinal fusion operations, where an inter-body cage is inserted to restore the disc space height and fuse the spinal segment [1, 2]. Among various surgical techniques for interbody cage insertion, oblique lumbar interbody fusion (OLIF) involves anterior insertion of interbody cage [1, 3, 4]. OLIF is known for its indirect decompression; restoration of disc space height and reduction of spondylolisthesis. Additionally, support at the anterior part of disc space helps restore lumbar lordosis [5]. For these purposes, an interbody cage is usually inserted into the anterior one-third of the disc space in OLIF [5].

The surgical procedures for conventional OLIF in combination with percutaneous pedicle screw fixation involve anterior insertion of an interbody cage in a lateral position first, followed by insertion of pedicle screws and rod system in a prone position (OLIF-con). However, inserting the interbody cage at one-third of the disc space is difficult for patients with spondylolisthesis because spinal segments are not aligned [1, 4, 5]. In addition, the reduction of spondylolisthesis may not be effective when the cage is inserted first due to tightly inserted interbody cage. The issues may be an inherent limitation with the conventional sequence of OLIF [3, 4].

The recent introduction of spinal navigation has enabled anterior and posterior surgical procedures to be performed in one lateral position (OLIF-one) [6, 7]. Consequently, the surgeon can move to either the anterior or posterior aspect of the patient, as both approaches can be used at the same time, allowing for reduction of spondylolisthesis prior to insertion of interbody cage. During OLIF-one, percutaneous pedicle screw fixation is first performed under the guidance of spinal navigation from the back. Anteriorly, the tight annulus and disc space are released during the removal of disc and restoration of disc height by serially inserting trial cages. Posteriorly, spondylolisthesis is reduced with the pedicle screw and rod system without limitation by tightly inserted interbody cage [2, 4, 7, 8]. After the reduction of spondylolisthesis, the cage can be inserted at the "sweet spot" in aligned spinal segment.

The primary objective of this study was to compare the reduction of spondylisthesis between OLIF-one and OLIF-con. The secondary objective was to compare the location of the cage, clinical outcomes, complication rate, and operation time between the two techniques.

## Methods

### Patients

A retrospective review of patients who underwent either OLIF-one or OLIF-con was approved by the institutional review board and patients consent was waived (IRB no. 2208-155-1354, 2101-080-1187, and 2023-05-068-001). The study included 72 consecutive patients who underwent surgery [9–11] from January 2017 to August 2022 at two different institutes [Institution 1 and Institution 2; blinded for review] and had spondylolisthesis grade I according to Meyerding's classification with slippage greater than 3mm [12]. The surgeries were performed by two different surgeons, with one surgeon operating at each institute. In [institution 1; blind for a review], only OLIF-con was performed before April 2022. OLIF-one was performed after the introduction of spinal navigation. Before performing OLIF-one, every surgeon had more than 100 cases of experience with OLIF-con. In [institution 2; blind for a review], only OLIF-con was performed. All data were anonymized, and this study strictly followed the regulations of privacy protection laws. The collection and analysis of data began in November 2022.

### Surgical techniques and hospital course

**OLIF-con.**  The standard surgical techniques were followed. After general anesthesia, the patient was placed in a 90-degree right lateral decubitus position. The anterior retroperitoneal approach was performed first, and the disc space was identified under the guidance of fluoroscope. The guidewire was inserted at the most desirable location of the disc space (anterior one-third of the disc, if possible), and serial dilators were introduced along the guidewire. A table-mounted self-retaining retractor was then applied. The disc space was cleaned using Cobb's elevator and curette, and then an interbody cage was inserted into the disc space after the insertion of serial trial cages.

The height and width of cages varied based on the size of the adjacent disc space, but the anterior-posterior length was the same (18mm) in all patients (Clydesdale® spinal system, Medtronic Sofamor Danek USA Inc., Memphis, TN, USA). After closure of the anterior incision, the patient's position was changed to prone, and a percutaneous pedicle screw system was applied posteriorly under fluoroscopic control. A specialized reducing device, rod reducer and screw extender, in the pedicle screw system (CD Horizon® Longitude® II fixation system, Medtronic Sofamor Danek USA Inc., Memphis, TN, USA) was used to reduce the spondylolisthesis, after which the pedicle screw and rod system was assembled with set screws.

**OLIF-one.**  The patient was placed in a 90-degree right lateral decubitus position, the same as for OLIF-con. The patient was located at the lateral margin of the table to facilitate the insertion of pedicle screws in a lateral position (Fig 1). The key distinction between OLIF-one and OLIF-con was the sequence of the surgical procedures. In OLIF-con, the sequence was anterior followed by posterior. However, in OLIF-one, the surgical procedures were performed in the sequence of posterior, anterior, posterior, anterior and then posterior again. The percutaneous pedicle screws were inserted posteriorly under spinal navigation guidance (O-arm™ Surgical Imaging Systems, Medtronic Sofamor Danek USA Inc., Memphis, TN, USA). After the insertion of pedicle screws, the anterior retroperitoneal approach was performed in the same way as for OLIF-con under spinal navigation guidance. The disc space was cleared using the same technique as for OLIF-con. After preparation of the cranial and caudal endplate, serial trial cages were inserted to restore disc height. The insertion of serial trial cages loosened the tight disc space and made the spinal segment mobile.

Then, the surgeon moved to the back of the patient, and used the same specialized reducing device as in OLIF-con to reduce the spondylolisthesis. The set screws were inserted but not

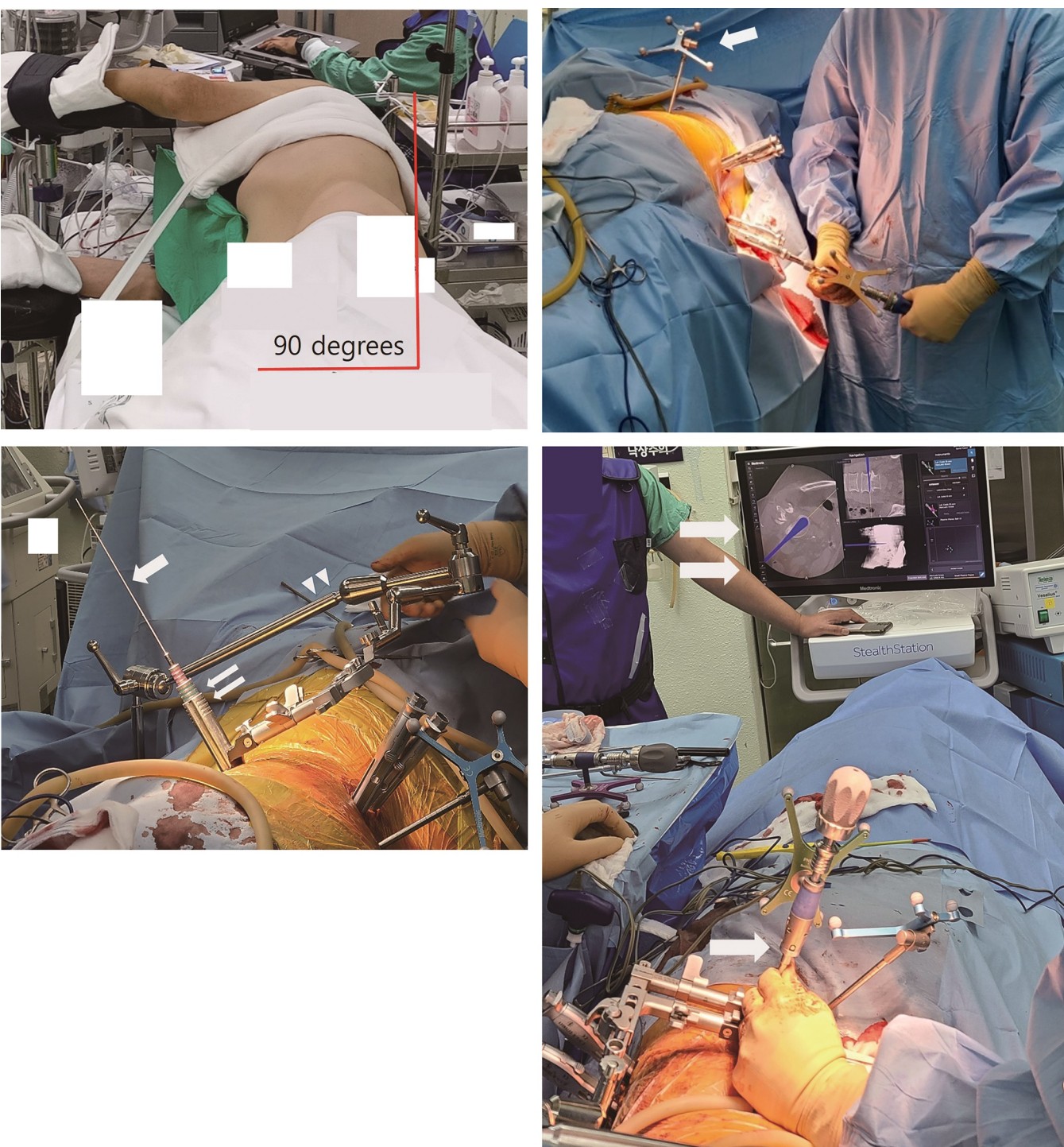

**Fig 1. Surgical procedures of oblique lumbar interbody fusion in one position.** (A) Patient position. The patient was positioned in a lateral decubitus position at a 90-degree angle at the posterior margin of the table. (B) Insertion of pedicle screws. The pedicle screws were percutaneously inserted under the guidance of surgical navigation (arrow). (C) Approach to the anterior disc space. A retroperitoneal approach is used to reach the anterior disc space. The guidewire (arrow) is inserted into the disc space and serial dilators (double arrows) are introduced along the guidewire. A table-mounted retractor (arrowheads) is then used to hook up a self-retaining retractor. (D) Preparation of the disc space. Under the guidance of surgical navigation (double arrow), the disc space is cleaned using a Cobb's elevator (arrow).

fully tightened to allow for sliding of the head of the pedicle screws along the rod during the reduction of spondylolisthesis and insertion of the interbody cage. Anteriorly, the anterior margins of the cranial and caudal vertebra were identified under surgical vision, and the disc space was cleaned again. Then, the interbody cage was inserted into the disc space, after which the pedicle screw and rod system was finally assembled with set screws.

**Hospital course.** Postoperatively, all patients were encouraged to ambulate from the day of surgery and were scheduled for discharge on postoperative day 3–4. Any postoperative events were recorded in the medical records.

## Data

The following data were reviewed: patients' demographic data (age, sex, body mass index [BMI, kg/m$^2$], and surgical level), intraoperative data (anesthesia time, surgical time from skin to skin, and injury to vessel, peritoneum, or bowel), complications (postoperative motor deficit (weakness on leg), sensory changes (hypesthesia), surgical site infection, ileus, or hematoma), and radiological outcomes (preoperative spondylolisthesis slippage [pre-slip, Fig 2A], postoperative spondylolisthesis slippage [post-slip, Fig 2B], reduction degree of spondylolisthesis [1-post-slip/pre-slip], and the location of the cage [location % from anterior margin of caudal vertebra to the center of cage, Fig 3]). The radiological measurements were performed by a radiologist who was blinded to the surgical method, using 200% magnified images. If the patient received multilevel fusion, the most severe spondylolisthesis level was measured. Clinical outcomes were assessed with the numeric rating scale (NRS) of pain in the back (NRS-back, /10), leg (NRS-leg, /10), and Oswestry disability index (ODI, %). The preoperative and postoperative 1-month outcomes were compared. Complications such as temporary motor and sensory deficit were also compared.

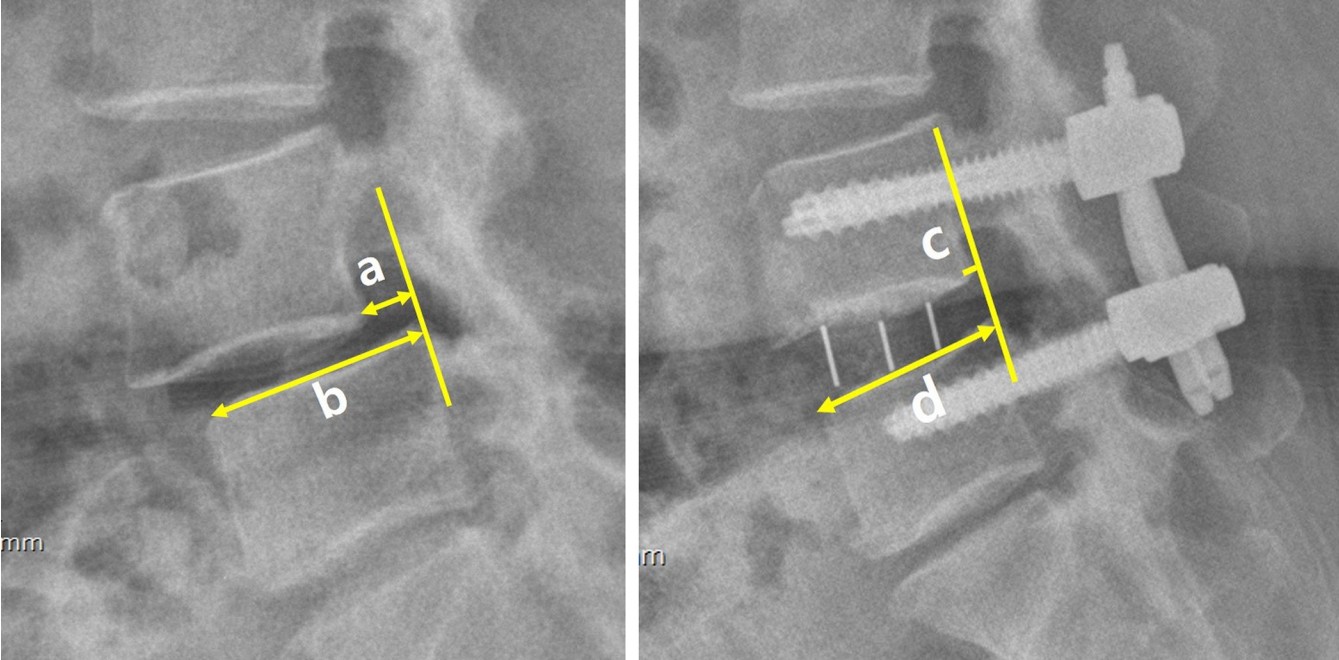

**Fig 2. Measurement of pre-and postoperative spondylolisthesis slippage.** (A) Measuring preoperative spondylolisthesis slippage (pre-slip = a/b). (a) is the horizontal distance from the posterior margin of the caudal vertebra to the upper vertebra and (b) is the length of the superior endplate of the caudal vertebra. (B) Measuring postoperative spondylolisthesis slippage (post-slip = c/d). (c) is the horizontal distance from the posterior margin of the caudal vertebra to the upper vertebra and (d) is the length of the superior endplate of the caudal vertebra.

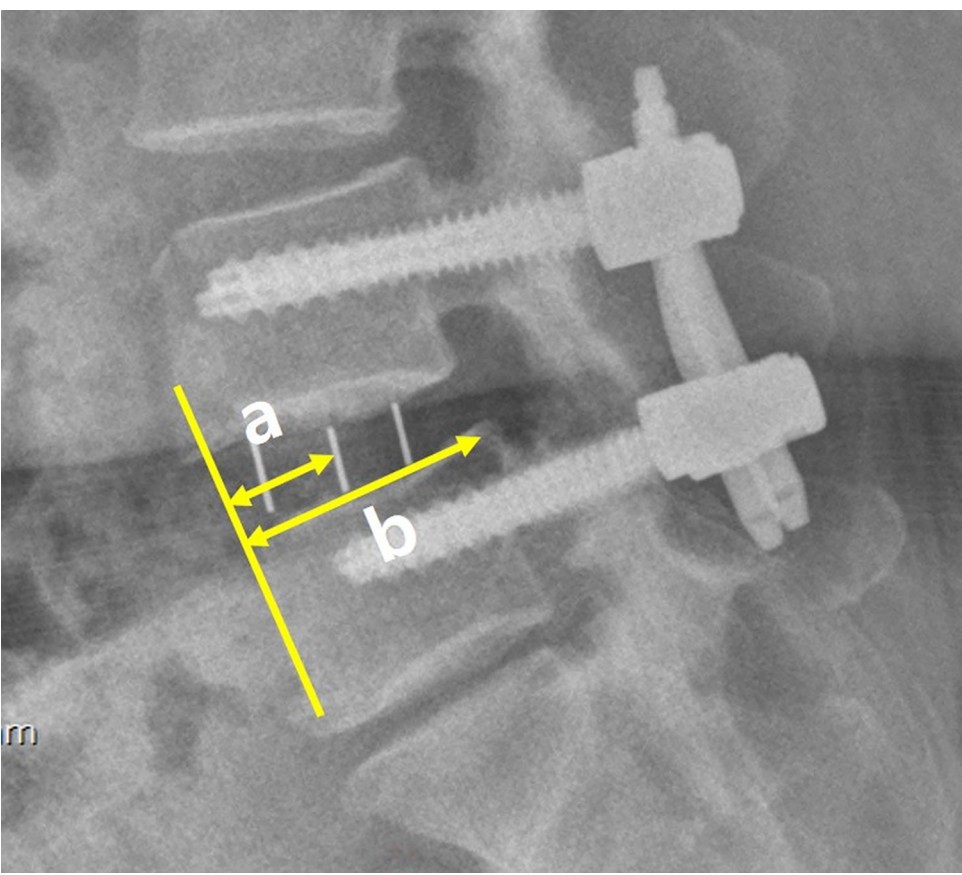

**Fig 3. Measurement of cage location.** Cage location is defined as a/b. (a) is the distance from the anterior margin of the caudal vertebra to the middle of the cage, while (b) is the length of the superior endplate of the caudal vertebra.

## Statistical analysis

All continuous data were presented as mean ± standard deviation, and non-continuous data were presented as number (proportion). Continuous parameters were compared using Student's t-test or Mann-Whitney U test, and non-continuous parameters were compared using the Chi-square test. All analyses were performed using SPSS® Statistics 26 (IBM Co., New York, USA). A two-tailed p-value of less than 0.05 was considered statistically significant.

## Results

### Baseline characteristics

Among 72 patients, 42 received OLIF-con (Male: Female = 32: 10; mean age, 66.5±7.4 years), and 30 received OLIF-one (Male: Female = 23: 7; mean age, 67.6±7.9 years). There was no significant difference in patients' demographics between the two groups (Table 1). Surgical time in OLIF-one (215 ± 42.8 minutes) was shorter than that in OLIF-con (268 ± 82.5 minutes), but the difference did not reach statistical significance (p = .10). Total anesthesia time also showed no significant difference between OLIF-con (348 ± 111 minutes) and OLIF-one (320 ± 62.5 minutes) (p = .32, Table 2).

### Radiologic and clinical outcome

Pre-slip was 23.1 ± 7.0% in OLIF-con and 26.3 ± 7.7% in OLIF-one (p = .15). Post-slip was 7.4 ± 5.8% in OLIF-con and 6.6 ± 6.2% in OLIF-one (p = .62). The reduction of slippage was

**Table 1. Patient demographics.**

|  | OLIF-con | OLIF-one | p-value |
|---|---|---|---|
| **Number of patients** | 42 | 30 | |
| **Age (years)** | 66.5±7.4 | 67.6±7.9 | 0.92 |
| **Sex (Male: Female)** | 32: 10 | 23: 7 | 0.96 |
| BMI (kg/m$^2$) | 25.6±3.1 | 24.2±3.1 | 0.21 |
| **Surgery level(s)** | | | 0.14 |
| One | 28 | 25 | |
| Two | 10 | 5 | |
| Three | 4 | 0 | |
| **Level of spondylolisthesis** | | | 0.63 |
| L2-3 | 5 | 6 | |
| L3-4 | 8 | 3 | |
| L4-5 | 20 | 15 | |
| L5-S1 | 9 | 6 | |
| **Preoperative NRS-B (/10)** | 7.2±2.4 | 7±2.2 | 0.76 |
| **Preoperative NRS-L (/10)** | 7.2±3.0 | 7.7±1.9 | 0.53 |
| **Preoperative ODI (%)** | 53.1±17.7 | 48.4±26.3 | 0.42 |

Abbreviations: BMI, body mass index; NRS-B, numeric rating scale of back; NRS-L, numeric rating scale of leg; ODI, Oswestry disability index; OLIF, oblique lumbar interbody fusion; OLIF-con, conventional oblique lumbar interbody fusion; OLIF-one, oblique lumbar interbody fusion in one position

65.4 ± 5.7% after OLIF-con and 74.4 ± 6.3% after OLIF-one (p = .04, Table 2 and Fig 4). The cage was located at 42.8 ± 10.3% after OLIF-con and 34.2 ± 8.9% after OLIF-one (p = .004). Postoperative transient weakness of the leg occurred in one patient each after OLIF-con and

**Table 2. Radiologic and clinical outcomes.**

|  | OLIF-con | OLIF-one | p-value* |
|---|---|---|---|
| **Reduction of slippage (%)** | 65.39±5.7 | 74.40±6.3 | 0.04* |
| **Pre-slip (%)** | 23.10±7.0 | 26.29±7.7 | 0.15 |
| **Post-slip (%)** | 7.39±5.8 | 6.56±6.2 | 0.62 |
| **Location of the cage (%)** | 42.8±10.3 | 34.2±8.9 | 0.004* |
| **Anesthesia time (minutes)** | 348±111 | 320±62.5 | 0.10 |
| **Surgical time (minutes)** | 268±82.5 | 215±42.8 | 0.32 |
| **Segmental angle (degrees)** | | | |
| Preoperative (degrees) | 6.17±8.05 | 9.17±10.85 | 0.26 |
| Postoperative (degrees) | 10.51±7.57 | 14.7±7.48 | 0.06 |
| **Postoperative 1-month NRS-B (/10)** | 3.6±5.5 | 3±5.3 | 0.47 |
| **Postoperative 1-month NRS-L (/10)** | 3.7±8.7 | 3.9±6.0 | 0.86 |
| **Postoperative 1-month ODI (%)** | 38.4±23.2 | 34.22±37.4 | 0.58 |
| **Postoperative motor weakness** | 1 | 1 | |
| **Postoperative sensory changes** | 2 | 1 | |

*p < .05; statistically significant
Abbreviations: NRS-B, numeric rating scale of back; NRS-L, numeric rating scale of leg; ODI, Oswestry disability index; OLIF, oblique lumbar interbody fusion; OLIF-con, conventional oblique lumbar interbody fusion; OLIF-one, oblique lumbar interbody fusion in one position; Post-slip, postoperative spondylolisthesis slippage; Pre-slip, preoperative spondylolisthesis slippage

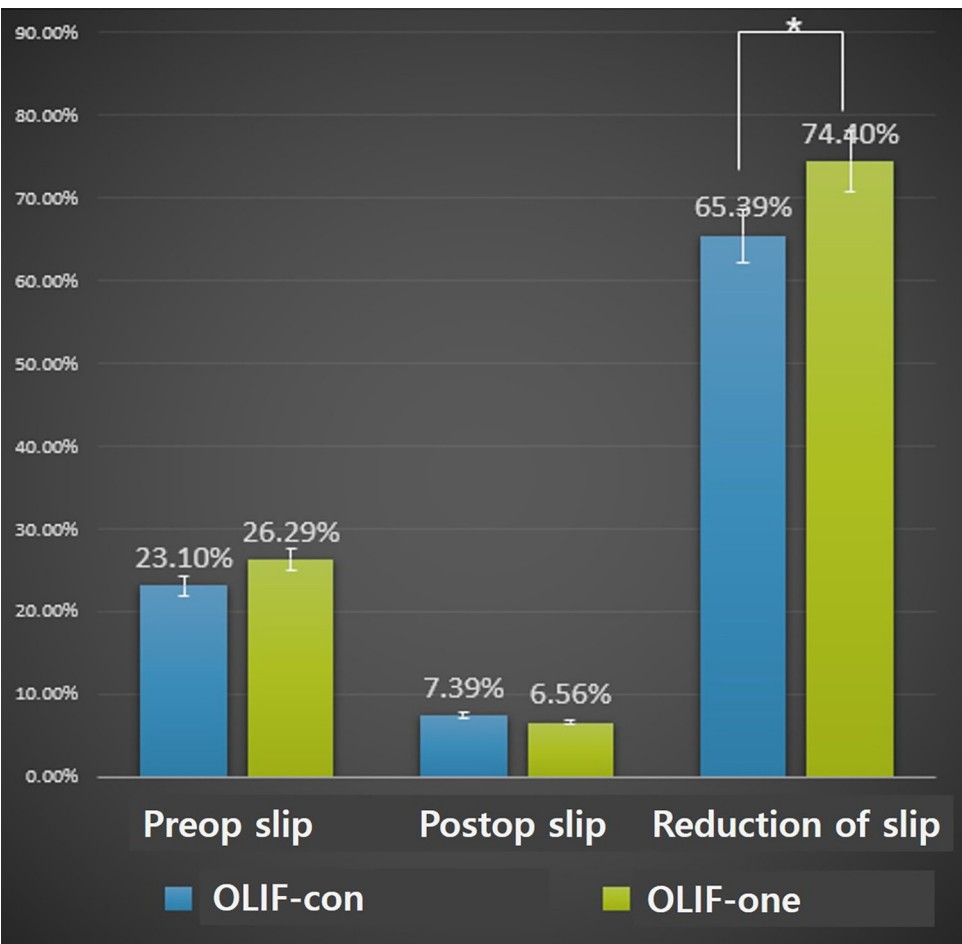

**Fig 4. Change of spondylolisthesis before and after surgery in two groups.** The bar graphs show the mean values and standard errors. An asterisk is used to indicate statistical significance (p = .04).

OLIF-one. Postoperative sensory changes on the leg occurred in two patients after OLIF-con and one after OLIF-one. These transient neurological problems corresponded to the nerve of the surgical level and recovered at postoperative 1-month. Other complications, such as surgical site infection, ileus, or hematoma, did not occur in either group. Postoperatively, clinical symptoms (NRS-B, NRS-L, and ODI) showed significant improvement, but there was no significant difference between the two groups (p > .05, Table 2).

## Discussion

The objective of this study was to compare the reduction of spondylolisthesis and the placement of cage between OLIF-one and OLIF-con. The study found that OLIF-one was associated with a significantly higher amount of spondylolisthesis reduction compared to OLIF-con. Although the location of the cage was not statistically significant between the two groups, OLIF-one tended to result in a more anterior placement of the cage than OLIF-con. Clinical outcomes and complications were not significantly different between the two groups. Although the study was conducted on a relatively small sample size, the findings suggest that changing the sequence of surgical procedures could facilitate the reduction of spondylolisthesis and the insertion of the cage at a desirable location.

## Treatment of spondylolisthesis

Lumbar spondylolisthesis can cause discomfort to patients due to spinal foraminal and/or central stenosis, and surgery may be recommended when non-surgical treatments fail to improve the symptoms [13–15]. We included patients with more than 3mm of anterior slip based on the findings of a prospective study that showed similar outcomes between decompression surgery and fusion surgery when the slip was less than 3mm [13, 14, 16]. The goals of fusion surgery for spondylolisthesis involve decompression, re-alignment of anterior vertebral slip, and correction of instability. While re-alignment of spondylolisthesis may not be a significant concern in cases of direct decompression and fusion surgery [2], it is crucial for effective indirect decompression, which is the core technique of OLIF [15, 16]. In this regard, we analyzed the effectiveness of reduction of spondylolisthesis as part of the initial surgical procedure (OLIF-one).

## Oblique lumbar interbody fusion

There are various surgical methods for treating spondylolisthesis, which can be broadly classified as ventral or dorsal approaches based on the direction of approach to the intervertebral disc space [3, 10, 17]. OLIF is a minimally invasive ventral approach that involves inserting a wide cage into the disc space [3, 15, 18–23]. Unlike the trans-psoas approach, surgeons approach the disc space anterior to the psoas muscle in OLIF to minimize the risk of injury to the muscle and lumbosacral nerve plexus [4]. The main concept of OLIF is indirect decompression, thus increasing the disc height and re-aligning spondylolisthesis are important surgical procedures [10].

Conventionally, OLIF surgical procedures were divided into ventral and dorsal procedures. In the former, the interbody cage was inserted from a ventral approach in a lateral decubitus position, while insertion of pedicle screw and reduction of spondylolisthesis were performed in a prone position as the latter procedure [10]. It is typically recommended to insert the interbody cage in the anterior one-third of the disc space to achieve effective lumbar lordosis and decompression [10]. However, locating the cage at the "sweet spot" may not be simple when the spinal segments are not aligned. The cage may be positioned centrally in the disc space when the caudal vertebra is used as a reference or outside the disc space when the cranial vertebra is used as a reference [1, 4, 5].

In addition to the location of the cage, the reduction of spondylolisthesis is also critical for effective indirect decompression [24]. However, reducing spondylolisthesis is limited by tightly inserted cages in OLIF-con. Thus, reduction of spondylolisthesis first without a tightly inserted interbody cage and insertion of the cage after the reduction would be ideal. However, it was not possible with the separated surgical procedures performed in different positions of the patient. Recently, the introduction of surgical navigation has enabled OLIF in one position, and switching between surgical procedures has become possible [7]. Switching freely between anterior and posterior surgical procedures enables appropriate reduction of spondylolisthesis and placement of the cage at the "sweet spot". In addition, performing the entire surgical procedure with the patient in a single position, specifically the lateral decubitus position, may reduce the influence of gravity, which can limit the reduction of spondylolisthesis.

## Limitations

This study had several limitations. Firstly, although patient data were collected from two different institutes, the sample size in this study was small, which may have increased the risk of type I or type II statistical errors. In addition, the study did not involve randomization of patients, and each patient was operated on by one of two individual surgeons. Moreover, the

selection of OLIF-con or OLIF-one was based on the availability of spinal navigation system, which could have introduced bias into the results. However, switching the sequence of surgical procedures was not a difficult task in OLIF-one, and it may facilitate reaching the surgical goal. Secondly, the desirable location of the cage may differ among patients based on the main problem. Therefore, locating the cage at the anterior one-third could not be generalized. Thirdly, this study did not assess long-term outcomes, and we did not consider radiologic data, such as spinal mobility, degree of spondylosis, facet joint arthrosis, and bone mineral density, which varied among individual patients. Also, the change of surgical sequence showed significant difference in radiologic outcomes between OLIF-con and OLIF-one, but did not show significant difference in clinical outcomes between the two groups. Future studies including longer follow-up period may also show significant difference in clinical outcomes as well. Fourthly, the cost-effectiveness of OLIF-one was not considered in this study. Surgical navigation is a costly surgical technique [25], and the efficacy of OLIF-one should be addressed in further studies. Nonetheless, despite these limitations, this study was meaningful in showing a solution in addressing the limitations of OLIF-con by switching surgical procedures, and it may improve the effectiveness of OLIF.

## Conclusion

This is the first study to demonstrate the superior outcomes of performing OLIF with supplementary pedicle screw fixation for patients with spondylolisthesis using a spinal navigation system in one lateral position. Moreover, this technique facilitated the reduction of spondylolisthesis and precise cage placement at the optimal position.

## Supporting information

**S1 Table. Data from [institution 1; blind for a review].**
(XLSX)

**S2 Table. Data from [institution 2; blind for a review].**
(XLSX)

## Acknowledgments

The authors appreciate the statistical advice provided by the Medical Research Collaborating Center at Seoul National University Hospital.

## Author Contributions

**Conceptualization:** Junghoon Han, Chang-Min Ha, Woon Tak Yuh, Jun-Hoe Kim, Tae-Shin Kim, Chun Kee Chung, Chi Heon Kim.

**Data curation:** Junghoon Han, Chang-Min Ha, Woon Tak Yuh, Young San Ko, Jun-Hoe Kim, Tae-Shin Kim, Asfandyar Khan, Chi Heon Kim.

**Formal analysis:** Junghoon Han, Chang-Min Ha, Woon Tak Yuh, Young San Ko, Chi Heon Kim.

**Funding acquisition:** Chi Heon Kim.

**Methodology:** Chang-Hyun Lee, Asfandyar Khan, Chun Kee Chung.

**Project administration:** Chi Heon Kim.

**Resources:** Junghoon Han.

**Supervision:** Chang-Hyun Lee, Sungjoon Lee, Sun-Ho Lee.

**Validation:** Chang-Min Ha, Sungjoon Lee, Sun-Ho Lee.

**Visualization:** Chang-Min Ha.

**Writing – original draft:** Junghoon Han.

**Writing – review & editing:** Chang-Min Ha, Chun Kee Chung, Chi Heon Kim.

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
