## [Decision Letter · Decision Letter 0]

13 Jun 2023

PONE-D-23-14072The sequence of oblique lumbar interbody fusion: reduction first or cage first?PLOS ONE

Dear Dr. Kim,

Thank you for submitting your manuscript to PLOS ONE. After careful consideration, we feel that it has merit but does not fully meet PLOS ONE’s publication criteria as it currently stands. Therefore, we invite you to submit a revised version of the manuscript that addresses the points raised during the review process.

General:

The title of the manuscript is not representing the core principle of the methodology (spinal navigation) and findings of the manuscript

Lines 28-30 & 61-63: Oblique lumbar interbody fusion (OLIF) procedures are divided  into anterior insertion of interbody cage first in lateral position and insertion of pedicle screw and  rod system in a prone position (OLIF-con). " I absolutely disagree with the authors. OLIF is an anterior interbody fusion performed through lateral approach involves an oblique channel between the prevertebral venous structures and the psoas muscle. Transpedicular fixation **is not a part of OLIF technique.** It is a supplementary ‘complementary” fixation that adds to the stability of the construct. This point is very important, makes the manuscript confusing at several sentences and should be corrected.

Introduction

.    Lines 59-60: “For these purposes, an interbody cage is usually inserted into the anterior one-third of the disc space in OLIF [5]”. The same meaning was repeated again in lines 63-64: “The location of the cage is crucial for restoring lumbar  lordosis and providing indirect decompression” This repetition should be omitted.

.    Lines 70-72: “Consequently, the surgeon can switch the sequences  of posterior and anterior surgical procedures in one position by moving the front and back of the patient” This sentence is unclear and confusing. I think it should be corrected to: “The surgeon can move to anterior or posterior aspect of the patient because both approaches could be used at the same time.”

.    Line 76: I think the term “inserted” should be better used instead of “located”

Methods

.    Lines 88-91: “All  patients with spondylolisthesis greater than 3mm were included without any exclusion criteria, as the purpose of the study was to compare immediate postoperative radiologic and clinical  outcomes”  What is the relation between inclusion of all  patients with spondylolisthesis greater than 3 mm and the purpose to study immediate post-operative radiologic and clinical  outcomes?

.    Lines 140-141: “Then, the surgeon moved to the back of the patient, and used the same specialized  reducing device as in OLIF-con, with an undersized trial cage, to reduce the spondylolisthesis.”  This sentence is  confusing. The surgeons have already inserted the trial cage through the lateral position then moved to the back of the patient to do reduction utilizing the reduction system. So, I think the words “with an undersized trial cage” should be deleted.

.    Lines 142- 143: “After the reduction of spondylolisthesis, the set screws were inserted but not broken off to allow  for sliding of the vertebra along the rod during the insertion of the interbody cage.” This sentence is confusing and unclear.

.    Results

.    Lines 190-192: “Surgical  time was not significantly different between OLIF-con (268 ± 82.5 minutes) and OLIF-one (215 ± 42.8 minutes) (p = .10).” Better to be changed to “Surgical  time was shorter in OLIF-one (215 ± 42.8 minutes) than in OLIF-con (268 ± 82.5 minutes), however the difference did not reach statistical significance (p = .10).” 

.    Table 1:

The statistical difference between the 2 groups as regards male/female ratio should be reported. I think this can be examined by 2x2 contiguency table. The same as regards the difference between the number of levels in the 2 groups utilizing MxN contiguncy tables.

.    Conclusion

.    Lines 300-302: “This is the first study to demonstrate the outcome of switching the procedures during  OLIF using surgical navigation. OLIF-one facilitated the reduction of spondylolisthesis and  precise cage placement at the optimal position. ” This sentence should be corrected to: “This is the first study to show superior results of performing OLIF with supplementary transpedicular fixation of spondylolisthesis in one lateral position utilizing spinal navigation system. Even more, this technique facilitated the reduction of spondylolisthesis and  precise cage placement at the optimal position”

We look forward to receiving your revised manuscript.

Kind regards,

Mohamed El-Sayed Abdel-Wanis, Ph.D.

Academic Editor

PLOS ONE

Journal Requirements:

"This study was supported by Seoul National University Hospital research fund (grant no. 04-2021-0540, CHK); and by the Doosan Yonkang Foundation (800-20210527, CHK)."

Reviewers' comments:

Reviewer's Responses to Questions

**Comments to the Author**

1. Is the manuscript technically sound, and do the data support the conclusions?

Reviewer #1: Partly

Reviewer #2: Yes

2. Has the statistical analysis been performed appropriately and rigorously? 

Reviewer #1: Yes

Reviewer #2: I Don't Know

3. Have the authors made all data underlying the findings in their manuscript fully available?

Reviewer #1: Yes

Reviewer #2: No

4. Is the manuscript presented in an intelligible fashion and written in standard English?

Reviewer #1: Yes

Reviewer #2: Yes

5. Review Comments to the Author

Reviewer #1: 1.Spondylolisthesis are heterogenous group based on etiology, disc height and degree of severity, please include those information as well

2.Please give information about detailed information about the temporary sensory and motor deficit

Reviewer #2: • Line 78: Term OLIF con used. This terminology probably explained in line 73. Kindly do it clearly. Similarly, it is confusing sequence at line 118.

• OLIF con is open procedure with front back sequence and OLIF one is MIS navigation procedure with back, front back sequence?

• Industry name mentioned on line 113. (CD Horizon® Longitude® II fixation system, Medtronic Sofamor Danek USA Inc., Memphis, TN, USA)

• On page 122 again industry name spinal navigation guidance (O-arm™ Surgical Imaging Systems,

• 123 Medtronic Sofamor Danek USA Inc., Memphis, TN, USA).

6. PLOS authors have the option to publish the peer review history of their article (what does this mean?). If published, this will include your full peer review and any attached files.

Reviewer #1: No

Reviewer #2: No

---

## [Author Response · Author response to Decision Letter 0]

21 Jul 2023

We appreciate your thorough review of our manuscript. In the attached file, you will find point-by-point responses to each comment provided.

---

## [Editor Report · Decision Letter 1]

14 Aug 2023

PONE-D-23-14072R1Efficacy of navigation-guided pedicle screw fixation followed by oblique lumbar interbody fusion for the reduction of spondylolisthesisPLOS ONE

Dear Dr. Chi Heon Kim,

Thank you for submitting your manuscript to PLOS ONE. After careful consideration, we feel that it has merit but does not fully meet PLOS ONE’s publication criteria as it currently stands. Therefore, we invite you to submit a revised version of the manuscript that addresses the points raised during the review process.

Thank you for addressing the reviewer and editorial comments. However, in my opinion, there is still improvement potentials.

We look forward to receiving your revised manuscript.

Kind regards,

Mohamed El-Sayed Abdel-Wanis, Ph.D.

Academic Editor

PLOS ONE

Journal Requirements:

Additional Editor Comments:

The authors addressed almost all the reviewers and editorial comments. However, as regards the title, I think still the title is not representing the core importance of the study. I would suggest the following title: SURGICAL TREATMENT OF SPONDYLOLITHESIS BY OLIF AND TRANSPEDICULAR SCREW FIXATION: COMPARISNON BETWEEN CONVENTIONAL DOUBLE POSITION VERSUS NAVIGATION- ASSISTED SINGLE LATERAL POSITION. Consequently, the short title should be also changed. I would suggest: single versus double position OLIF and transpedicular screw fixation
---

## [Author Response · Author response to Decision Letter 1]

15 Aug 2023

We appreciate your thorough review of our manuscript. Below, you will find point-by-point responses to each comment provided by the reviewers and the editor in the attached file.

---

## [Editor Report · Decision Letter 2]

23 Aug 2023

Surgical treatment of spondylolisthesis by oblique lumbar interbody fusion and transpedicular screw fixation: comparison between conventional double position versus navigation-assisted single lateral position

PONE-D-23-14072R2

Dear Dr. Chi Heon Kim 

We’re pleased to inform you that your manuscript has been judged scientifically suitable for publication and will be formally accepted for publication once it meets all outstanding technical requirements.

Kind regards,

Mohamed El-Sayed Abdel-Wanis, Ph.D.

Academic Editor

PLOS ONE
---

## [Editor Report · Acceptance letter]

6 Sep 2023

PONE-D-23-14072R2 

Surgical treatment of spondylolisthesis by oblique lumbar interbody fusion and transpedicular screw fixation: comparison between conventional double position versus navigation-assisted single lateral position 

Dear Dr. Kim:

I'm pleased to inform you that your manuscript has been deemed suitable for publication in PLOS ONE. Congratulations! Your manuscript is now with our production department. 

Kind regards, 

on behalf of

Prof. Dr Mohamed El-Sayed Abdel-Wanis 

Academic Editor

PLOS ONE